# Size–curvature constraint in the closing motion of Venus flytrap leaves

**Michiko Hirata[1], Zichen Kang[1]¤, Hiroki Asakawa[2], Hiraku Suda[2]\*, Masatsugu Toyota[2,3,4], Yuji Ohashi[1], Satoru Tsugawa[1]\*¤**

**1** Department of Mechanical Engineering, Faculty of Systems Science and Technology, Akita Prefectural University, Yurihonjo, Akita, Japan, **2** Department of Biochemistry and Molecular Biology, Saitama University, Saitama, Japan, **3** Suntory Rising Stars Encouragement Program in Life Sciences (SunRiSE), Suntory Foundation for Life Sciences, Soraku-gun, Kyoto, Japan, **4** College of Plant Science and Technology, Huazhong Agricultural University, Wuhan, Hubei, China

¤Current address: Faculty of Engineering, Hokkaido University, Sapporo, Hokkaido, Japan
* suda222@mail.saitama-u.ac.jp (HS); tsugawa@eng.hokudai.ac.jp (ST)

## Abstract

Among carnivorous plants, the Venus flytrap (*Dionaea muscipula*) is known for its rapid (<1 s) trap closure. Although buckling instability, hydrostatic pressure, and hydroelastic coupling have all been proposed to be involved, the nature of this process and the relationship between trap size and curvature remain elusive. Here, we monitored the closure of Venus flytraps and performed micro–CT scanning and 3D reconstruction, revealing that increasing angular velocity was correlated with higher values of a non-dimensional shape index. Based on these experimental data, we constructed a geometric model of the trap that takes leaf orientation into account. We found that leaf curvature is dependent on leaf size, a relationship we denote as a size–curvature constraint. We further propose a curvature design derived from differential deformations of a two-layer model of the leaf, which could be a powerful tool to control the curvatures of soft and bending surface structures in the field of biomimetics.

## Introduction

Charles Darwin described the Venus flytrap as "one of the most wonderful plants in the world," perhaps more because of the rapid closing (within 1 s) of its component leaves, which is initiated by a slight touch stimulus, than because of its function of catching insects [1]. This fast closure typically occurs when Venus flytrap leaves receive a second stimulation within 30 s after the first, which can look to observers as if the plants are waiting and watching the insect [2,3]. Given that the plant body does not have an equivalent to animal muscles, how Venus flytraps achieve high closure speeds has been a subject of research and debate [4–6]. Also of interest is how the plants, lacking a nervous system, transmit touch-induced signals; the regulation

**Data availability statement:** All data files and related rendering files are available from the github (https://satorutsugawa.github.io/flytrap_geometric_model_datashare/).

**Funding:** This work was supported by Japan Society for the Promotion of Science (JSPS) KAKENHI grant numbers JP23H01143, JP22J00902, JP25KJ0714, JP24H00565, JP25K18499, JP25K18427, JST CREST grant number JPMJCR2121, and JST ERATO grant number JPMJER2403. The funders had no role in study design, data collection and analysis, decision to publish, or preparation of the manuscript.

**Competing interests:** The authors declare no competing interests.

of $Ca^{2+}$ concentrations and action potentials have received attention in this regard [7–12]. Unresolved questions include whether there is a condition that controls the ability of the leaves to close and whether closure is accelerated by some form of stress release.

Early experiments demonstrated the trap's ability to close and the conditions under which this occurs. Darwin observed that the open state is very stable: the trap does not close spontaneously, even under the influence of raindrops and wind gusts [1,13]. The flytrap can close both in air and under water conditions [14]. In addition, seedling traps <1 cm in length can close, but their closure time (also called snapping time) is > 5 s, longer than typical times for adult traps [14]. Uncouplers and blockers of membrane channels inhibit trap closure, making closing very slow [15]. Trap closure is not triggered by sustained displacement of a sensory hair or by deflection that is sufficiently slow, and only one touch could initiate the motion of the leaves if sensory hair deflection is above a threshold [9]. These results indicate that fast closure requires certain conditions: leaf maturity, rapid deflection of the sensory hair, internal biological processes that occur after stimulation, and appropriate regulation of membrane channels.

The outer surface of the trap expands after closure [3,4,16] and the plastic extensibility of the outer surface increases [17], indicating that the crucial factors in fast closure may relate to properties of the outer surface. Therefore, it is tempting to think that the leaves undergo an elastic buckling mechanism, as proposed previously, resulting in a smooth snapping transition from the open to the closed state [4,18]. However, we showed that traps lacking outward curvature also close, and the stretching energy and the curvature energy of the outer surface increase simultaneously, thereby suggesting the possibility of an additional effect separate from the elastic snap–buckling instability [19].

A recently proposed model, the hydrostatic pressure coupling model [15,16,20], posits that the curvatures of the multiple layers of leaves composing a trap may be driven by the independent turgor pressures of the layers. The changes in the trap might thus reflect the time scales of membrane channel opening and subsequent water flow. However, in this model the closure speed does not depend on the trap shape, and thus the model does not take into account the experimental data showing higher closure speeds for larger traps [4]. In the context of plant tropism, there appears to be a geometric relationship between curvature of the material elements and differential growth rates [21]. Therefore, we reasoned that a basic geometric model might explain the relationship between the shape and the deformational effects of the multiple layers.

In this study, we experimentally monitored the closure of Venus flytraps and quantified their non–dimensional geometric parameters, finding a trend relating the shape and the closing speed. We then built geometrical models of the open and closed states of the trap based on micro–CT scanning followed by 3D reconstruction. Analyzing these data, we determined that the trap closure is governed by a size–curvature constraint. We propose that the trap features a curvature-controlling design that underlies trap motion.

## Materials and methods

### Plant material and imaging conditions

Venus flytrap plants were purchased from Hanadonya Associe in Japan and cultivated in the laboratory with a natural light environment. For three-dimensional quantification of trap closure, two cameras (DC-G9L-K and DC-GH6L, Panasonic). The cameras were located at stereo angles of about 45° and –45° toward the specimen. Calibration was performed using a calibrator of a precisely fabricated cuboid (30 mm × 30 mm × 8 mm) with two slits (6 mm × 3 mm × 30 mm). The recording speed was 60 fps. Trap closure was stimulated by careful human touch manipulation. ImageJ multi-point tool tracking was used to detect and track the feature points. Surface mesh construction from the point cloud was performed using Meshlab software.

### Micro–CT

A Venus flytrap trap leaf was cut at the petiole, and micro–CT data were acquired using a Skyscan1276 (Bruker, USA) under the following conditions: resolution, 1008 × 672 pixels; source voltage, 40 kV; source current, 200 μA; image pixel size, 30.000639 μm; depth, 16 bits; exposure, 90 ms; rotation step, 0.600 degree; frame averaging, 2. Data were obtained from the same trap leaf in both the open and closed states. Three-dimensional reconstructions from the CT imaging data were generated using NRecon (Bruker, USA) and InVesalius (Centro de Tecnologia da Informação Renato Archer, Brazil). The reconstructed models were then converted into solid objects using Meshmixer (Autodesk, USA).

### Elliptic Fourier transformation

We applied elliptic Fourier transformation to the 21 cross-sectional contour data. The contour data contains $n$-th number of $x$-coordinates $x_n$ and $y$-coordinates $y_n$ of the cross-sectional contour. We calculated the Fourier series expansion for $x_n$ and for $y_n$ independently.

The $x$- and $y$-coordinates can be rewritten as

$$x(t) = \sum_{n=1}^{N} \left[ A_n \cos\left(\frac{2\pi nt}{T}\right) + B_n \sin\left(\frac{2\pi nt}{T}\right) \right], \ y(t) = \sum_{n=1}^{N} \left[ C_n \cos\left(\frac{2\pi nt}{T}\right) + D_n \sin\left(\frac{2\pi nt}{T}\right) \right]$$

where $n$ is the harmonic number, $N$ is the maximum harmonic number, $t$ is the displacement along the contour, and $T$ is the total displacement. The elliptic Fourier coefficients are

$$A_n = \frac{T}{2n^2\pi^2} \sum_{p=1}^{k} \frac{\Delta x_p}{\Delta t_p} \left[ \cos\left(\frac{2\pi nt_p}{T}\right) - \cos\left(\frac{2\pi nt_{p-1}}{T}\right) \right], \ B_n = \frac{T}{2n^2\pi^2} \sum_{p=1}^{k} \frac{\Delta x_p}{\Delta t_p} \left[ \cos\left(\frac{2\pi nt_p}{T}\right) - \cos\left(\frac{2\pi nt_{p-1}}{T}\right) \right]$$

$$C_n = \frac{T}{2n^2\pi^2} \sum_{p=1}^{k} \frac{\Delta y_p}{\Delta t_p} \left[ \cos\left(\frac{2\pi nt_p}{T}\right) - \cos\left(\frac{2\pi nt_{p-1}}{T}\right) \right], \ D_n = \frac{T}{2n^2\pi^2} \sum_{p=1}^{k} \frac{\Delta y_p}{\Delta t_p} \left[ \cos\left(\frac{2\pi nt_p}{T}\right) - \cos\left(\frac{2\pi nt_{p-1}}{T}\right) \right]$$

where $k$ is the total number of steps around the contour, $\Delta x$ and $\Delta y$ are the displacements along the $x$-axis and the $y$-axis between points $p$ and $p + 1$, $\Delta t$ is the length of the step between points $p$ and $p + 1$, and $t_p$ is the accumulated length of step segments at point $p$. We used about 200 data points along the contour and performed the elliptic Fourier transformation with $n = 40$ harmonics.

### 3D reconstruction

We used a 3D reconstruction method, direct linear transformation (DLT) based on the detailed description in [20].

## Calculation of the elastic strain energy

We performed elastic energy estimation based on the detailed description in ref. [20].

## Results and discussion

### The speed of trap closure increases with trap size

To assess the relationship between Venus flytrap trap shape and speed, we experimentally observed the closing motions of traps of various sizes and shapes ($n$ = 15). We measured the width $W$, and height $H$ (Fig 1a) and the leaf opening angle $\theta$ (Fig 1c). $\theta$ was measured as the angle between the lines connecting the leaf joint and the base of the trap teeth (Supplementary S1 Fig). The 95% confidence interval for $W$ was 10.8–16.5 (mm), and that for angle was 0.77–1.15 (rad). We observed that traps with width <6 mm or >21 mm did not move, possibly due to leaf immaturity or senescence, respectively (Fig 1b) [14]. Furthermore, the positive angular velocity $\omega_{angle}$ = $(\theta_{open} - \theta_{closed})/2\Delta t$ was distributed within a certain

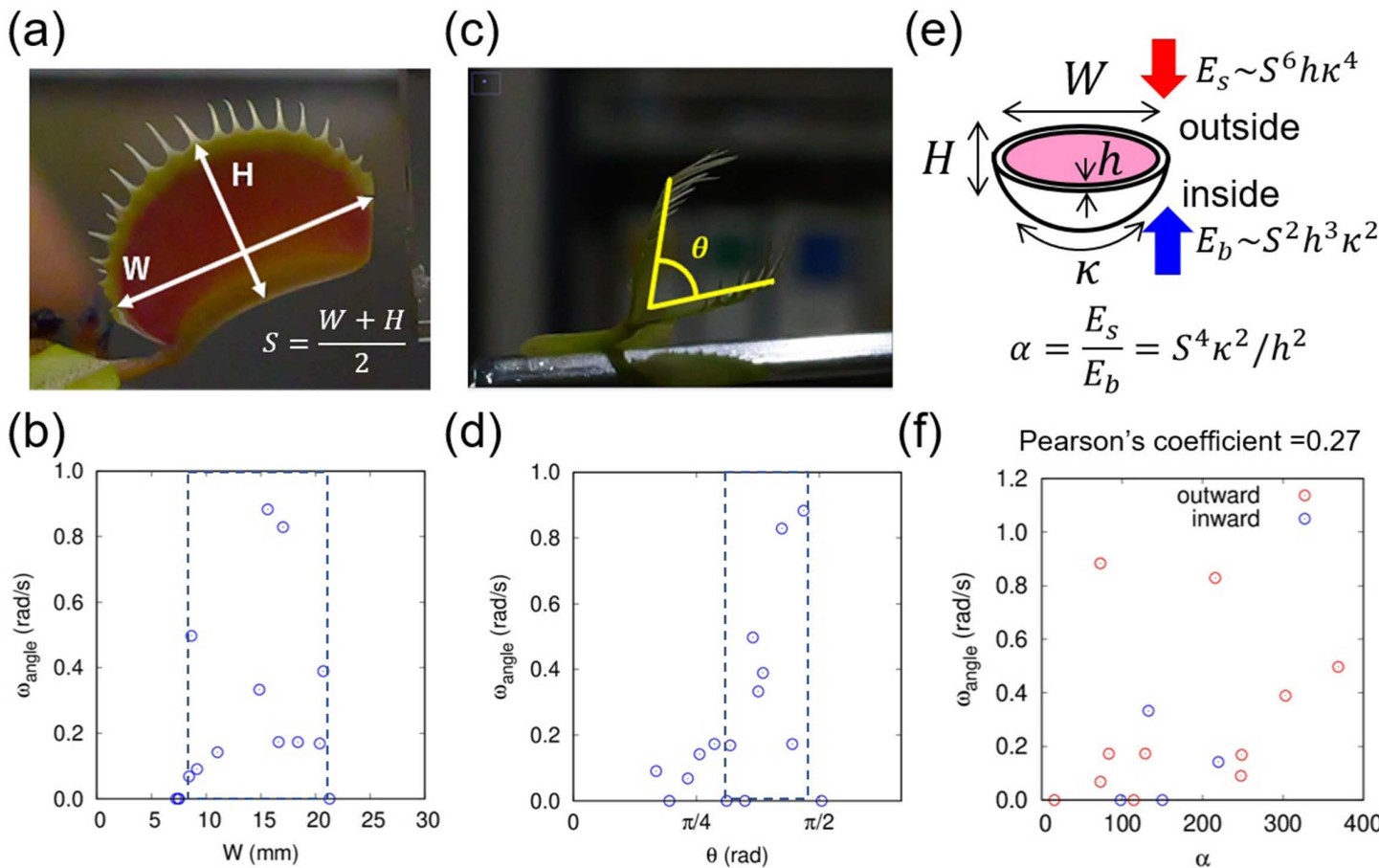

**Fig 1. Relationships between trap speed and size, leaf angle, and stretching–bending ratio. (a)** Definitions of trap width **W**, height **H** and size **S**. **(b)** Leaf angular velocity versus leaf width. **(c)** Definition of leaf angle θ. **(d)** Leaf angular velocity versus leaf angle. **(e)** Schematic defining the stretching–bending ratio. The stretching energy—the energy causing the outside of the leaf to flip outward and become flattened—is denoted by the red vector arrow and the bending energy—the energy causing the inside of the leaf to resist that outward pull and remain curved—by the blue vector arrow. **(f)** Leaf angular velocity ($\omega_{angle}$) versus stretching–bending ratio ($\alpha$). Outwardly and inwardly curved leaves are plotted in red and blue, respectively. The p-value of the increasing trend was 0.00373.

range of leaf widths (defined by the dotted rectangle in Fig 1b). We detected an increasing trend in leaf angular velocity with greater initial leaf angle (Fig 1d); notably, traps with $\omega_{angle}$ near $\pi/2$ rad closed rapidly (dotted rectangle). To visualize this trend in a size-independent fashion, we adopted a non-dimensional geometric parameter, the stretching–bending ratio $\alpha = S^4 \kappa^2 / h^2$, (where $S = (W + H)/2$), which is the ratio between the stretching energy flattening the leaf flat and the bending energy resisting that flattening ([4]; Fig 1e). Here, $S/h$ and $S\kappa$ represent size–thickness anisotropy and size–warping anisotropy, respectively; the curvature $\kappa$ was measured at the leaf midline (Supplementary S2 Fig). The speed of closure increased as the shape index (stretching–bending ratio) increased (Fig 1f), consistent with the relationship observed in previous work [4]. These results indicate that rapid closure requires a certain size and angle.

The dimensionless index $\alpha$ was adopted as an effective parameter to capture the balance between thickness and elastic modulus. In the context of the flytrap, which consists of hydrated, multilayered tissues, $\alpha$ can be interpreted as reflecting the mechanical response of a bilayer system in which differential strain – potentially arising from turgor, water transport, and cell wall extensibility – drives bending while being constrained by in-plane stretching resistance. Thus, $\alpha$ effectively incorporates the combined influence of tissue hydration, layer thickness, and cell wall mechanics into a single dimensionless parameter.

## Geometric parameters for open and closed trap states can be derived from micro–CT scanning data

Next, to understand the trap shape more clearly, we implemented the following previously published geometric model [18]:

$$\begin{pmatrix} x_0 \\ y_0 \\ z_0 \end{pmatrix} = \begin{pmatrix} \sin(\alpha)\left[R_0 + \beta H \cos(d\alpha)\right] \\ 2D\beta(1-\beta)\cos\left(\frac{\pi}{2}\frac{\alpha}{\alpha_{max}}\right) \\ R_0\cos(\alpha) - R_0 + \beta H \cos(\alpha)\cos(d\alpha) \end{pmatrix}$$

(1)

Here, $R_0$ is the radius of the midrib, $H$ is the leaf height, $d$ is the decreasing degree of height along the midrib, $\alpha_{max}$ is the degree of leaf waving along the midrib, and $D$ is the degree of leaf bending (i.e., curvature in the height axis). Taking the rotation of the leaf around $x$-axis at the midrib into account, we introduced an orientational parameter $\phi$, which is the rotation angle around the $x$-axis with the origin at the midrib.

$$\begin{pmatrix} x \\ y \\ z \end{pmatrix} = \begin{pmatrix} x_0 \\ y_0\cos\phi - \left(z_0 - R\cos\alpha + R\right)\sin\phi \\ y_0\sin\phi + \left(z_0 - R\cos\alpha + R\right)\cos\phi + R\cos(\alpha) - R \end{pmatrix}$$

(2)

To extract these geometric parameters, we used micro–CT scanning data fitted to the parameters. From the data for the open (Fig 2a) and closed states (Fig 2b), 21 cross–sectional contours were interpolated by elliptic Fourier transformation (Methods, Fig 2c). Using the interpolations, we aligned the 3D coordinates of the open and closed states (Fig 2d). We estimated $R_0$ from the circle fitting of the midrib, measured $H$ as the leaf height, and estimated $d$ as the difference between the height at the center and the height at edges of the trap. $\alpha_{max}$ was estimated to be $70\pi/180$ rad. Using these estimates, we obtained the following parameters:

Open: $R_0 = 5.1$mm, $H = 6.0$mm, $d = 1.0$, $D = 0.0$mm, $\alpha_{max} = \frac{70\pi}{180}$ rad, $\phi = -\frac{\pi}{4}$ rad

Closed: $R_0 = 5.1$mm, $H = 6.0$mm, $d = 1.0$, $D = 2.6$mm, $\alpha_{max} = \frac{70\pi}{180}$ rad, $\phi = 0$ rad

These results indicated that the parameter $D$ is crucial for distinguishing the open and closed states and that the inclination angle $\phi$, which denotes the declination angle of the whole leaf, also differs between the open and closed states.

Assuming that the leaf sizes are the same, the essential parameter D represents the curvature along the leaf perpendicular to the midrib and serves as an geometric descriptor of bending during trap closure. A smaller D indicates a flattened, open state, whereas a larger D corresponds to a strongly curved, closed state. Physiologically, D reflects the differential deformation between the inner and outer tissue layers derived from curvature, driven by asymmetries including

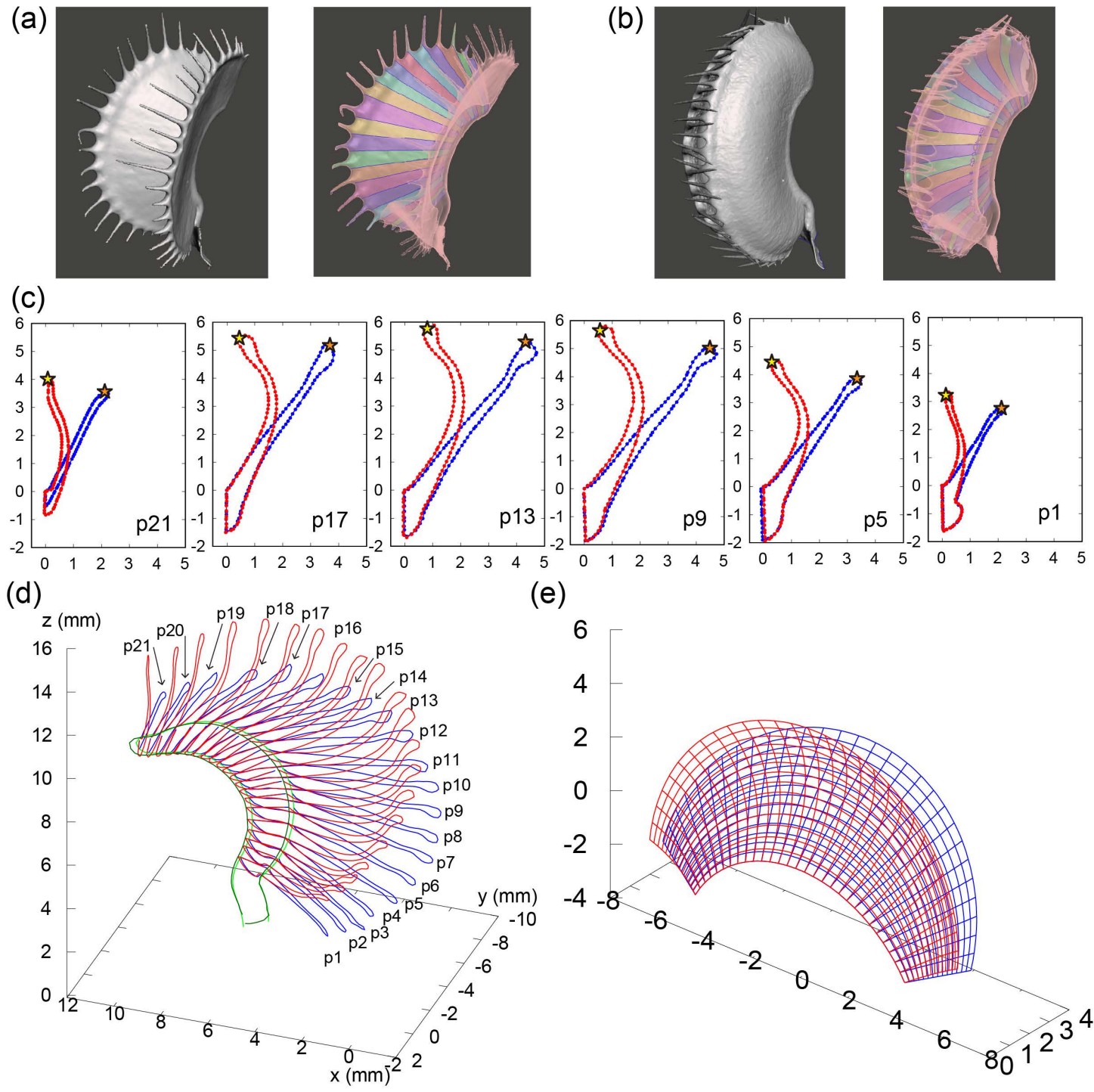

**Fig 2. Acquisition of geometric parameters from traps in open and closed states based on micro–CT scanning. (a, b)** 3D segmentation (left) and divided cross-sections (right) of the open (a) and closed states **(b)**. **(c)** Cross-sectional data interpolated by elliptic Fourier transformation for the open (blue) and closed states (red). We defined the cross-sections p1, p2, …, p21 from the petiole to the top of the leaf along the midrib. Stars indicate the points corresponding to the leaf tips. **(d)** Reconstructed leaves in open (blue) and closed states (red). **(e)** Geometric models of the open (blue) and closed states (red).

those in turgor pressure, water transport, and cell wall mechanical properties. From a biophysical perspective, D can be interpreted as the cumulative effect of strain differences between the two layers translated by curvature change, arising from differential expansion rates. One possible factor underlying these differences is ion fluxes, such as $Ca^{2+}$-mediated signaling, which may control water movement across membranes and generate rapid volume changes. Thus, D could reflect the coupling between cellular-scale physiological processes and organ-scale mechanical behavior. Further perturbation experiments will clarify the link between the parameters (D and φ) and their physiological meanings, which is currently indirect.

### *D* and φ are significant parameters for the closing motion

We performed further experiments to reconstruct the 3D deformation dynamics. Using two cameras located to the left and right of the flytraps, we took movies of their closing motions (Methods, Fig 3a, Supplementary S1 Movie). We marked the

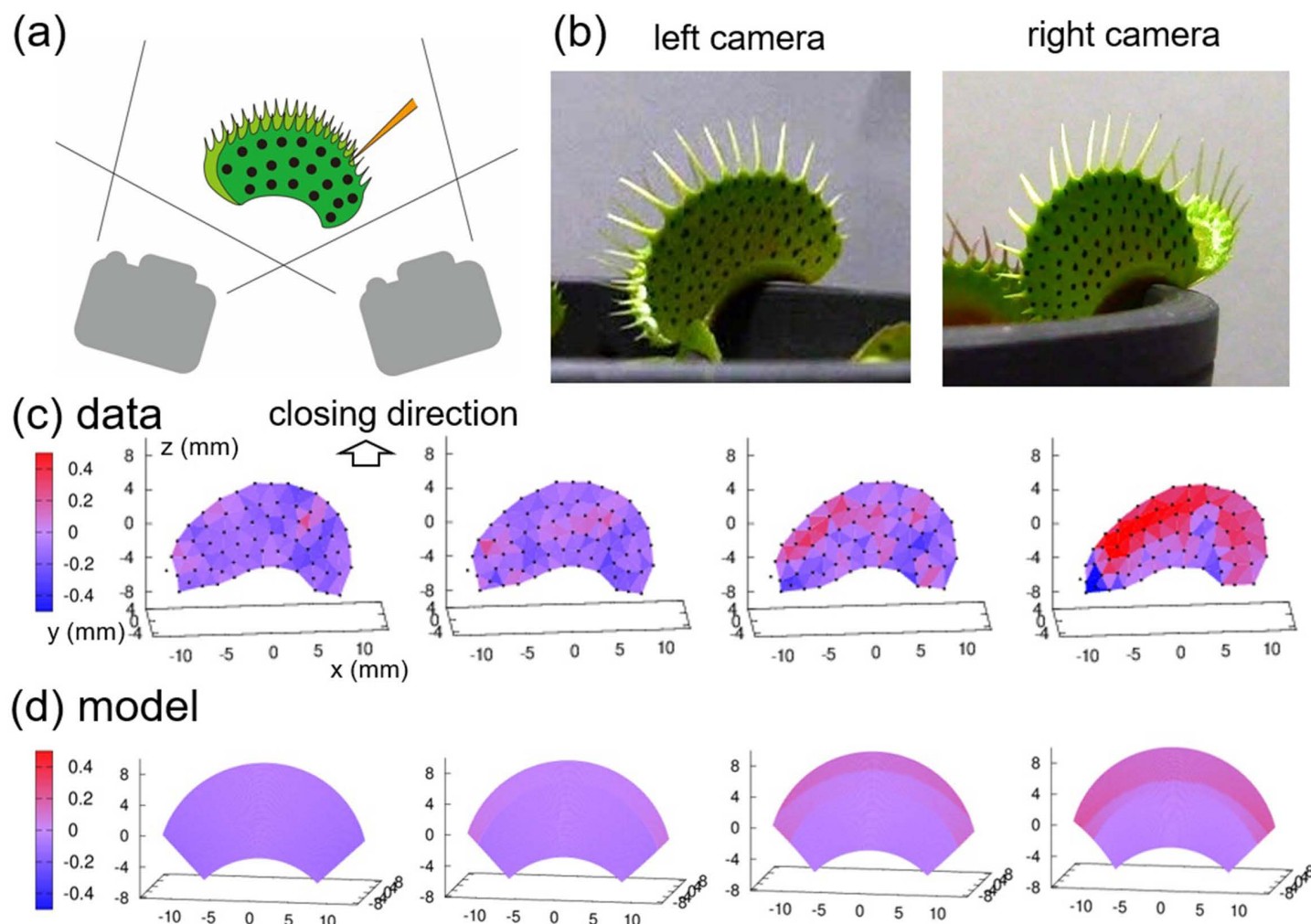

**Fig 3. Acquisition of geometric parameters over time using a 3D reconstruction method. (a)** Illustration of the two angles of observation with markers on the outer surface of the trap. **(b)** Snapshots of the initial state obtained from the left and right cameras. **(c)** Spatiotemporal mean curvature of the 3D reconstructed data. The color code refers to mean curvature. **(d)** Spatiotemporal mean curvature of the reconstructed geometric model with fitted geometric parameters.

outer surface of each trap with characteristic black dots using a marker pen and tracked the dots using ImageJ (Fig 3b). This enabled us to quantify the spatiotemporal dynamics of the mean curvature of the traps (Fig 3c).

We then estimated the geometric parameters as follows. We measured $H = 16.0$ mm based on the distance between top and bottom points and $d = 1.10$ based on the curved boundary of the leaf top. We calculated the curvature radius $R = 10.3$ mm using ImageJ. The ranges of $\alpha$ and $\beta$ were determined to be –45 deg to 45 deg and 0 to 0.78, respectively. We estimated the leaf angle at the initial state $\phi = 0.40$ rad from the initial leaf angle. The remaining parameter $D = 6.93$ mm was inferred from the closed morphology. Using these parameters, we were able to reconstruct a geometric model of the spatiotemporal change of the leaf (Fig 3d), which showed large changes in the peripheral region that are consistent with the micro–CT scanning data (Fig 2d).

In summary, we showed that $D$ and $\phi$ alone are sufficient to reproduce the movement in our simulations, and thus significant parameters for the closing motion, using a 3D reconstruction method.

## Trap closure is subject to a size–curvature constraint

Based on the actual data set, we could evaluate the closing motion from only parameters $D$ and $\phi$. Since $\phi$ represents the rotation of the coordinate system, we fixed parameter $\phi$ first and assessed the closing motion in a morphospace that characterized the morphological changes using a small number of parameters, i.e., $H$ and $D$ (Fig 4a). In the morphospace, the closing motion can be described by the changes in location from one point to another (vectors). To quantitatively assess the morphology clearly, we demonstrated the mean curvature over time corresponding to the vectors (Fig 4b). Our results showed that the mean curvature increases as the leaf height $H$ decreases, as expected based on the definitions of those parameters (Fig 4b). Here, the important point is that the curvature $D$ changes within 0.5 s from the open state (when $D_{op} = 0.0$ mm) to the closed state ($D_{cl} = 0.7$ mm), and there is a correspondence between curvature and time in this geometric model.

To confirm the actual data corresponding to $D$, we calculated the parameters $D_{op}$ and $D_{cl}$ from the measured values for the leaf curvature radius $R_l$, $H$, and $\phi$ using the following geometric relationships between those two parameters and the equation $R_l = 1/\kappa$ (Fig 4c),

$$D_{op} \simeq R_l - \sqrt{R_l^2 - \left(\frac{H}{2}\right)^2} \tag{3}$$

$$D_{cl} = \frac{H}{2}\tan\phi \tag{4}$$

We then evaluated the angular velocity with respect to curvature $\omega_{curv}$ as,

$$\omega_{curv} = \frac{(D_{cl} - D_{op})\beta\left(1 - \frac{\beta}{2}\right)}{H\Delta t}, \quad \beta = 0.5 \tag{5}$$

This estimated angular velocity $\omega_{curv}$ was relatively lower than the observed angular velocity in Fig 1 (Fig 4d). Precisely speaking, the angular velocity shown in Fig 1 was measured as the angle difference between the open and closed states, whereas the angular velocity in eq. (5) was measured from the difference in curvature between them. The angle difference reflects the global change of whole curved surface whereas the curvature difference reflects the local change of waving structure of the leaf. As the angular velocity in terms of angle $\omega_{angle}$ is comparable to the closing speed of the whole leaf structure, $\omega_{angle}$ is better to represent the whole closing motion where the leaves experience both the angle change and curvature change simultaneously. In addition, we found that the measured $D_{cl}$ increased as a function of $H$, even though

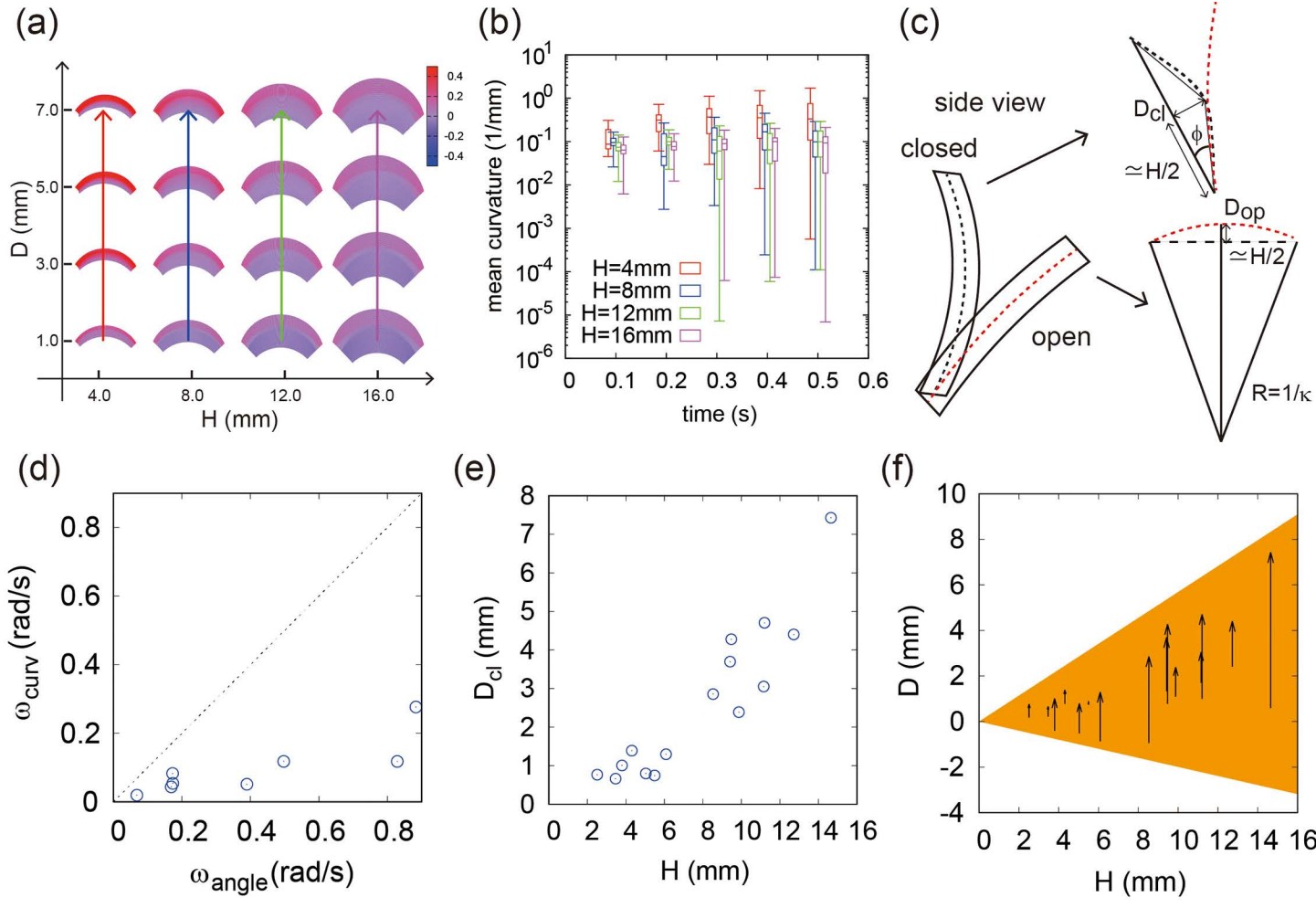

**Fig 4. Morphospace analysis and the relationship between trap shape and speed. (a)** Morphospace in terms of the height **H** and the deflection **D**. The color code refers to mean curvature. **(b)** Temporal change in the mean curvatures of the four models in **(a)**. Boxplot shows the spatial average of the mean curvatures for each model. **(c)** Schematic illustration of the indices $D_{cl}$ and $D_{op}$. **(d)** Comparison between the angular velocity with respect to curvature, $\omega_{curv}$, and that with respect to angle, $\omega_{angle}$. **(e)** Relationship of index $D_{cl}$ and height **H**. **(f)** Relationship of **D** to height **H**. The arrows denote the vectors from $D_{op}$ to $D_{cl}$. The ratio **D** to **H** is constrained within a limited range (orange area).

$\phi$ can be variable in eq. (4) (Fig 4e). Thus, the morphospace of *H* and *D* with actual data reveals the existence of a size–curvature constraint (orange area in Fig 4f), as all of the data are found within a certain range of the ratio *D/H*. This implies that small traps cannot bend faster than larger traps.

In summary, we discovered that the effect of $\omega_{angle}$ is dominant in the closing motion compared to that of $\omega_{curv}$ (Fig 4d). In addition, we confirmed that there is a size–curvature constraint in the closing motion (Fig 4f).

In our observations, the sign of curvature reverses during motion in many leaves, suggesting a contribution of elastic instability (Fig 4f). However, we also found several cases in which initially flat surfaces bent during closure (Fig 4f), suggesting that even leaves lacking elastic energy storage can sometimes exhibit leaf movement. These exceptional motions may instead be accounted for by the hydrostatic pressure model [15,16,20]. When considered within a multilayer framework, this model offers an informative basis for interpreting differential deformation. These mechanisms likely provide the underlying driving forces are manifested at the organ scale. Our findings suggest that size-dependent geometric

constraints may be applicable to any of these existing models and will be essential for understanding of the trap mechanics. An important direction for future work is to integrate turgor-driven deformation and elastic instability within a size-aware framework and experimentally test their relative contributions.

**Size-dependent curvature derives from differential deformations of different layers**

As indicated by the results described in previous sections, trap shape and size are subject to certain constraints, making it possible to construct a curvature formula based on differential deformation dynamics. In doing so, we use a two–layer model of the trap (with inner and outer layers) and denote their expansion rates as $\dot{\varepsilon}_{in}$ and $\dot{\varepsilon}_{out}$, respectively. The thickness of the layers is denoted $h$. The temporal derivative of the curvature in the height direction $\kappa$ can be written as follows (Fig 5a, see also [21]).

$$\frac{\partial \kappa(t)}{\partial t} = \frac{1}{h}\left(1 - \kappa(t)^2 h^2\right) E(t)\Delta(t)$$

$$\dot{E}(t) = \frac{\dot{\varepsilon}_{in} + \dot{\varepsilon}_{out}}{2}$$

$$\Delta(t) = \frac{\dot{\varepsilon}_{out} - \dot{\varepsilon}_{in}}{\dot{\varepsilon}_{in} + \dot{\varepsilon}_{out}}$$

Considering that the outer surface expands after trap closure ([3,4,16,19]), the outer deformation is expected to be significantly larger than the inner deformation, i.e., $\dot{\varepsilon}_{out}(t) \gg \dot{\varepsilon}_{in}(t)$. Therefore,

$$\frac{\partial \kappa(t)}{\partial t} = \frac{1}{2h}\left(1 - \kappa(t)^2 h^2\right) \dot{\varepsilon}_{out}(t)$$

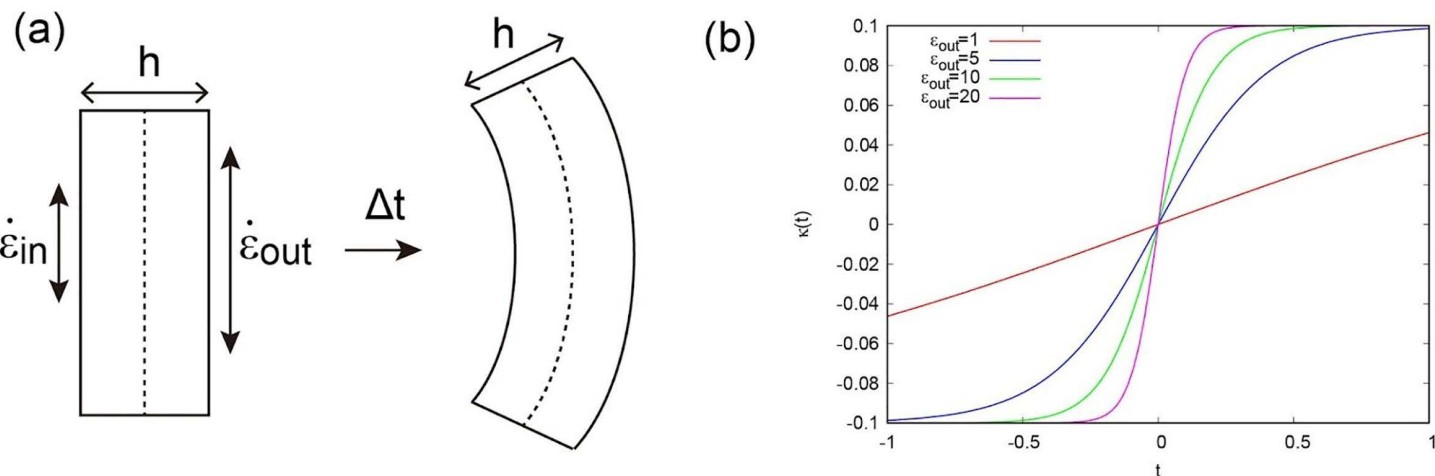

**Fig 5. Dynamic relationship between curvature and strain rate.** (a) Illustrations of the two-layer model. (b) Simulated results of κ(t)~tanh(t) with different $\dot{\varepsilon}_{out}$.

Assuming that $\dot{\varepsilon}_{out}(t)$ is constant ($\dot{\varepsilon}_{out}$) during the period of rapid closure, we can solve the equation, and the solution becomes,

$$\kappa(t) = \frac{1}{h}\tanh\left[0.5\dot{\varepsilon}_{out}\left(t-t_1\right)\right]$$

where $\kappa(t)$ becomes 0 at $t = t_1$. To capture the closure behavior, we set $t_1 = 0$ and assess the temporal change in curvature. We calculated the temporal dependence of $\kappa(t)$ for different $\dot{\varepsilon}_{out}$, showing an increase of the change in curvature as $\dot{\varepsilon}_{out}$ increases (Fig 5b).

Our model still has several limitations. First, it assumes spatially uniform deformation and therefore does not capture the heterogeneous strain distribution observed across the leaf surface. Second, the model still did not reflect the complex tissue architecture of the trap such as the midrib, marginal teeth, and hinge region, which may play key roles in guiding and stabilizing trap closure. For future studies, it is important to incorporate cell wall mechanics, such as nonlinear elasticity and extensibility, nor potential viscoelastic effects that may contribute to time-dependent deformation during rapid motion. As different parts of the leaf may respond at distinct time scale during closure [22], it is essential to include the regional differences in temporal dynamics. Incorporating these biological and mechanical complexities would improve the realism of the model and make a model accurate representation of the underlying mechanisms governing trap movement.

## Conclusions

Consistent with previous studies such as [4] and [14], size-dependent movement has previously been proposed, but the specific geometric factors constraining trap motion have remained unclear. Our results suggest the presence of a size–curvature constraint and further indicate that traps smaller than 6 mm were unable to deform, and traps larger than 21 mm reached a limit preventing them from deforming fully. These bounds may reflect a balance between driving forces, such as turgor-induced differential strain, and geometric or mechanical constraints including bending stiffness. While this interpretation remains qualitative, it provides a potential physical basis for the observed morphospace. Predicting these limits from first principles may facilitate the biological interpretation of the underlying physical constraints. We did not examine cellular-scale dynamics; however, investigating spatial and temporal variations in cell size, shape, and deformation may help clarify how these multiscale processes contribute to size dependent trap closure.

## Supporting information

**S1 Fig. Three examples of measuring changes in leaf opening angles from the open state to the closed state.** The opening angle was measured as the angle of the lines connecting the leaf joint and the base of the teeth.
(JPG)

**S2 Fig. Eight examples of measuring the curvatures of the midlines of the leaves.** Yellow points indicate points detected along the midline.
(JPG)

**S1 Movie. Two videos showing the closing movement of a Venus flytrap, filmed from two different angles.**
(GIF)

## Author contributions

**Conceptualization:** Yuji Ohashi, Satoru Tsugawa.

**Data curation:** Michiko Hirata, Zichen Kang, Hiroki Asakawa, Hiraku Suda, Masatsugu Toyota.

**Formal analysis:** Zichen Kang, Satoru Tsugawa.

**Funding acquisition:** Satoru Tsugawa.

**Investigation:** Michiko Hirata, Satoru Tsugawa.

**Methodology:** Michiko Hirata.

**Supervision:** Masatsugu Toyota, Yuji Ohashi, Satoru Tsugawa.

**Validation:** Michiko Hirata, Satoru Tsugawa.

**Visualization:** Michiko Hirata, Satoru Tsugawa.

**Writing – original draft:** Satoru Tsugawa.

**Writing – review & editing:** Hiraku Suda, Satoru Tsugawa.

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
