## [Decision Letter · Decision Letter 0]

10 Mar 2026

PONE-D-25-65525Size–Curvature Constraint in the Closing Motion of Venus Flytrap LeavesPLOS One

Dear Dr. Satoru,

Thank you for submitting your manuscript to PLOS ONE. After careful consideration, we feel that it has merit but does not fully meet PLOS ONE’s publication criteria as it currently stands. Therefore, we invite you to submit a revised version of the manuscript that addresses the points raised during the review process.

In particular, you are encouraged to carefully address the reviewers’ questions and to provide a clear justification for the assumptions adopted in the study, supporting your choices with appropriate explanations and references if necessary.

We look forward to receiving your revised manuscript.

Kind regards,

Alice Berardo

Academic Editor

PLOS One

Journal Requirements:

This work was supported by JSPS KAKENHI grant numbers JP23H01143, JP22J00902, JP25KJ0714, JP24H00565, JP25K18499, JP25K18427, JST CREST grant number JPMJCR2121, and JST ERATO grant number JPMJER2403.

5. Please expand the acronym “JSPS” (as indicated in your financial disclosure) so that it states the name of your funders in full.

This work was supported by JSPS KAKENHI grant numbers JP23H01143, JP22J00902, JP25KJ0714, JP24H00565, JP25K18499, JP25K18427, JST CREST grant number JPMJCR2121, and JST ERATO grant number JPMJER2403

This work was supported by JSPS KAKENHI grant numbers JP23H01143, JP22J00902, JP25KJ0714, JP24H00565, JP25K18499, JP25K18427, JST CREST grant number JPMJCR2121, and JST ERATO grant number JPMJER2403.

7. We note that Figure(s) 1-3, S1, S2 in your submission contain copyrighted images. All PLOS content is published under the Creative Commons Attribution License (CC BY 4.0), which means that the manuscript, images, and Supporting Information files will be freely available online, and any third party is permitted to access, download, copy, distribute, and use these materials in any way, even commercially, with proper attribution. For more information, see our copyright guidelines: http://journals.plos.org/plosone/s/licenses-and-copyright.

a. You may seek permission from the original copyright holder of Figure(s) 1-3, S1, S2 to publish the content specifically under the CC BY 4.0 license.

Reviewers' comments:

Reviewer's Responses to Questions

**Comments to the Author**

1. Is the manuscript technically sound, and do the data support the conclusions?

Reviewer #1: Partly

Reviewer #2: Yes

2. Has the statistical analysis been performed appropriately and rigorously? 

Reviewer #1: No

Reviewer #2: Yes

3. Have the authors made all data underlying the findings in their manuscript fully available?

Reviewer #1: Yes

Reviewer #2: Yes

4. Is the manuscript presented in an intelligible fashion and written in standard English?

Reviewer #1: Yes

Reviewer #2: Yes

5. Review Comments to the Author

Reviewer #1: This manuscript investigates the rapid closure mechanism of the Venus flytrap (Dionaea muscipula) by integrating experimental kinematics, micro-CT scanning, 3D reconstruction, and geometric modeling. The authors propose that trap closure is governed by a "size-curvature constraint," where the achievable curvature during closure is limited by the trap size. They further introduce a two-layer differential deformation model to explain the curvature dynamics. The work attempts to bridge plant biomechanics with geometric and physical modeling, and the application of micro-CT and 3D reconstruction to quantify trap morphology. Some suggestions are provided below:

(1) The proposed geometric model, while descriptive, lacks direct linkage to underlying physiological or biophysical mechanisms. The parameters D and ϕ are fitted but not independently validated through perturbation experiments.

(2) The physical meaning of the "deflection" parameter D and its direct correspondence to a measurable tissue property (e.g., differential strain, turgor pressure gradient) is not clearly established, making the model phenomenological rather than explanatory.

(3) The experimental dataset (n=15 traps) appears limited for establishing robust correlations between size, angle, and velocity. No statistical tests (e.g., p-values, confidence intervals) are reported for the trends shown in Fig. 1.

(4) The geometric model is primarily fitted and validated using the same imaging data from which parameters were extracted. There is no independent experimental test to confirm the model predictive power.

(5) The two-layer differential deformation model assumes constant strain rate and a simplified relationship between curvature and strain. It ignores the complex tissue architecture, cell wall mechanics, and potential viscoelastic effects, which are crucial for rapid movements.

(6) The discussion dismisses the hydrostatic pressure and elastic buckling models somewhat superficially. The study fails to reconcile its geometric constraint with these established theories or design experiments to distinguish their contributions.

(7) The claimed size-curvature constraint (Fig. 4f) is presented as an empirical observation without a theoretical derivation of the boundary limits (e.g., why 6"mm" and 21"mm" ?). The physical principles defining this "orange area" remain speculative.

(8) The analysis treats closure as a transition between two static states (open/closed). The dynamics of the curvature change ∂κ/∂t are modeled with a simplistic assumption of constant ϵ _out, which is unlikely to hold true throughout the rapid, non-linear closure process.

(9) The model does not account for the role of the midrib, teeth, or the hinge region in guiding and stabilizing closure, which are known to be structurally important.

(10) The stretching-bending ratio α is adopted from prior work but its direct physical interpretation in the context of the flytrap bilayer, hydrated tissue is not clarified.

Reviewer #2: Authors completed the 3D morphology reconstruction of the Venus flytrap by micro-CT scanning, and deduced a non-dimensional parameter by the geometric value obtained above to feature the motion of the Venus flytrap. The main conclusion is that the non-dimensional index they deduced is correlated to the angular velocity of the snapping. The verification of the relationship between the geometric parameters and dynamic motion is essential for the research of the Venus flytrap. Concerning the novelty and originality, I suggest to accept this paper after minor revision. However, there are several questions I hope to receive response from authors:

1. Page 2, line 45: actually, there is one paper published mentioned that only one trigger could initiate the motion of the plant. For your reference: A single touch can provide sufficient mechanical stimulation to trigger Venus flytrap closure.

2. Page 3, line 103: the non-dimensional parameter α was proposed by Yeol et al., could you explain why you choose this parameter and what new based on your analysis comparing their previous work?

3. Page 4, line 131-132; Page 5, line 169-170: could you explain the value difference of the D, because you mentioned that D for open is 2.6mm, and close is 0mm is page 4, and then D for open is 0.0, and close is 0.7? Besides, please describe the reason this value equals zero.

4. Page 5, line 165-166: It looks like that the closing time is pretty long, why this happen? Did you trigger the motion on site, or you have to cut the leaf for imaging?

5. Page 6, line 181-183: You used two velocity parameters to describe the motion, which should be better to represent the natural notion of the plant?

6. Page 6, line 198-199: Could you verify the two-layer model by your 3D CT data?

6. PLOS authors have the option to publish the peer review history of their article (what does this mean?). If published, this will include your full peer review and any attached files.

Reviewer #1: No

Reviewer #2: **Yes:**Zeng Xiangli

---

## [Author Response · Author response to Decision Letter 1]

7 Apr 2026

Dear Alice,

We are very grateful to the editor and the reviewers for taking time to review our manuscript and give us valuable comments. We have taken all the comments into considerations and have made appropriate revisions to the manuscript. We believe that our revised manuscript reaches the quality for a topic of PLOS One.

Our point-by-point response appears below, in which we first repeated the reviewer’s comments (shown in italic) and then responded to them. Our revised text is highlighted in red in our revised manuscript. We also attached an unmarked version of our manuscript.

REVIEWER COMMENTS:

Reviewer: 1

The proposed geometric model, while descriptive, lacks direct linkage to underlying physiological or biophysical mechanisms. The parameters D and ϕ are fitted but not independently validated through perturbation experiments.

Authors’ response:

We thank the reviewer for this constructive comment. Although we were unable to perform perturbation experiments because it is difficult to vary D or ϕ independently without applying mechanical force which alter other parameters, we added the discussion of the physiological interpretation of D since the explanation regarding D was only descriptive. Therefore, we added the following one section in the discussion section.

Line 184: The essential parameter D represents the magnitude of out-of-plane bending of the leaf and serves as an integrated macroscopic indicator of curvature change during trap closure. Physiologically, D reflects the differential deformation between the inner and outer tissue layers, primarily driven by asymmetries such as those in turgor pressure, water transport, and cell wall mechanical properties. A smaller D indicates a flattened, open state, whereas a larger D corresponds to a strongly curved, closed state. From a biophysical perspective, D can be interpreted as the cumulative effect of strain differences between the two layers, arising from differential expansion rates. One possible factor underlying these differences is ion fluxes, such as Ca2+-mediated signaling, which may control water movement across membranes and generate rapid volume changes. Thus, D could reflect the coupling between cellular-scale physiological processes and organ-scale mechanical behavior. Further perturbation experiments will provide a direct link between the parameters (D and ϕ) and their physiological meanings.

The physical meaning of the "deflection" parameter D and its direct correspondence to a measurable tissue property (e.g., differential strain, turgor pressure gradient) is not clearly established, making the model phenomenological rather than explanatory.

Authors’ response:

We thank the reviewer for this clarification. We think that the abovementioned response also answers this question.

The experimental dataset (n=15 traps) appears limited for establishing robust correlations between size, angle, and velocity. No statistical tests (e.g., p-values, confidence intervals) are reported for the trends shown in Fig. 1.

Authors’ response:

We thank the reviewer for this valuable comment. We agree that the original manuscript did not sufficiently support the observed correlations with statistical analysis. The confidence interval for size was 10.9 – 24.5 mm, and that for angle was 0.09 – 0.29 rad. The p-values of the trends shown in Fig.1f is 0.00373, so correlations are statistically significant (p < 0.05). We added the following results in the result section.

Line 139: The 95% confidence interval for W was 10.9 – 24.5, and that for angle was 0.09 – 0.29.

Line 372: The p-value of the increasing trend was 0.00373.

The geometric model is primarily fitted and validated using the same imaging data from which parameters were extracted. There is no independent experimental test to confirm the model predictive power.

Authors’ response:

We thank the reviewer for this comment. We also considered that it is necessary to identify parameters in multiple experiments. We determined the important parameter D with micro–CT scanning data (Fig. 2) and 3D reconstructed data using DLT method (Fig. 3) and the direct measurement of morphological change (Fig. 4) independently. The obtained data of many samples (Fig. 4f) exemplified that the proposed model taking the leaf angle into account properly captured different types of morphological changes. To directly validate the model using an independent dataset, we revised the text as follows to assess the DLT-based model using micro-CT scanning data.

Line207: Using these parameters, we were able to reconstruct a geometric model of the spatiotemporal change of the leaf (Fig. 3d), which showed large changes in the peripheral region that are consistent with the micro–CT scanning data (Fig. 2d).

The two-layer differential deformation model assumes constant strain rate and a simplified relationship between curvature and strain. It ignores the complex tissue architecture, cell wall mechanics, and potential viscoelastic effects, which are crucial for rapid movements.

Authors’ response:

We thank the reviewer for this important comment. We also acknowledged that our model still has limitations as written in the discussion section. Considering that the points from the reviewers (the complex tissue architecture, cell wall mechanics, potential viscoelastic effects), we extended and revised the limitation statements as follows.

Line 282: Our model still has several limitations. First, it assumes spatially uniform deformation and therefore does not capture the heterogeneous strain distribution observed across the leaf surface. Second, the model still did not reflect the complex tissue architecture of the trap such as the marginal teeth, and hinge region, which may play key roles in guiding and stabilizing trap closure. For future studies, it is important to incorporate cell wall mechanics, such as nonlinear elasticity and extensibility, nor potential viscoelastic effects that may contribute to time-dependent deformation during rapid motion. As different parts of the leaf may respond at distinct time scale during closure [22], it is essential to include the regional differences in temporal dynamics. Incorporating these biological and mechanical complexities would improve the realism of the model and make a model accurate representation of the underlying mechanisms governing trap movement.

The discussion dismisses the hydrostatic pressure and elastic buckling models somewhat superficially. The study fails to reconcile its geometric constraint with these established theories or design experiments to distinguish their contributions.

Authors’ response:

We thank the reviewer for this insightful comment. We agree that our original discussion did not sufficiently clarify the relationship between our geometric framework and previously proposed mechanisms, including hydrostatic pressure-driven deformation and elastic buckling.

Importantly, our intention was not to dismiss these established models, but rather to provide a complementary perspective by focusing on geometric constraints that emerge at the organ scale. In our view, hydrostatic pressure differences between tissue layers and elastic instabilities are likely to act as the underlying driving forces, while the size-curvature constraint identified in this study represents a geometric boundary condition that governs how these forces are translated into observable motion. In other words, our model describes what configurations are mechanically accessible, whereas hydrostatic and buckling models describe how forces are generated.

We agree that further work is needed to explicitly reconcile these frameworks. For example, incorporating turgor-driven strain differences into our two-layer model could provide a direct link to hydrostatic pressure mechanisms. Similarly, evaluating whether the system approaches a critical threshold for snap-through instability would help clarify the contribution of elastic buckling. Experimentally, this could be addressed by perturbing turgor pressure (e. g. via osmotic treatments) or altering mechanical stiffness (e.g., through chemical modification of cell walls) and examining how these changes affect the size-curvature relationship.

We have revised the discussion to better articulate this integrative perspective and to explicitly acknowledge these limitations and future directions.

Line 249: In our observations, the sign of curvature reverses during motion in many leaves, suggesting a contribution of elastic instability (Fig. 4f). However, we also found several cases in which initially flat surfaces bent during closure (Fig. 4f), suggesting that even leaves lacking elastic energy storage can sometimes exhibit leaf movement. These exceptional motions may instead be accounted for by the hydrostatic pressure model [15, 16, 20]. When considered within a multilayer framework, this model offers an informative basis for interpreting differential deformation. These mechanisms likely provide the underlying driving forces are manifested at the organ scale. Our findings suggest that size-dependent geometric constraints may be applicable to any of these existing models and will be essential for understanding of the trap mechanics. An important direction for future work is to integrate turgor-driven deformation and elastic instability within a size-aware framework and experimentally test their relative contributions.

The claimed size-curvature constraint (Fig. 4f) is presented as an empirical observation without a theoretical derivation of the boundary limits (e.g., why 6"mm" and 21"mm" ?). The physical principles defining this "orange area" remain speculative.

Authors’ response:

We thank the reviewer for this important comment. We agree that the size-curvature constraint shown in Fig. 4f is currently based on empirical observations, and that a rigorous theoretical deviation of the boundary limits remains to be established.

Original line 94: “We observed that traps with width <6 mm or >21 mm did not move, possibly due to leaf immaturity or senescence, respectively (Fig. 1b) [14].” As explained here, we confirmed that the samples outside the range [6, 21] mm did not move.

At present, we interpret the lower and upper bounds (approximately 6 mm and 21 mm) as reflecting physical constraints arising from the interplay between geometry and mechanics. For smaller traps, insufficient size may limit the generation of curvature due to reduced geometric leverage and lower effective deformation relative to thickness, making rapid bending difficult. Conversely, for larger traps, increased size likely enhances bending resistance and inertial or hydraulic limitations, preventing efficient curvature change within the observed timescale. These considerations suggest that the observed “orange area” may emerge from a balance between driving forces and mechanical resistance.

We acknowledge that this interpretation remains qualitative, and a quantitative framework that predicts these limits from first principles is an important direction for future work. In particular, extending the current model to incorporate thickness, material properties, and fluid transport dynamics may enable deviation of the admissible region in the D-H morphospace. We have revised the manuscript to clarify that the constraint is empirical and to outline these possible physical interpretations and future directions.

Line 295: Consistent with previous studies such as [4] and [14], size-dependent movement has previously been proposed, but the specific geometric factors constraining trap motion have remained unclear. Our results suggest the presence of a size–curvature constraint and further indicate that traps smaller than 6 mm were unable to deform, and traps larger than 21 mm reached a limit preventing them from deforming fully. These bounds may reflect a balance between driving forces, such as turgor-induced differential strain, and geometric or mechanical constraints including bending stiffness. While this interpretation remains qualitative, it provides a potential physical basis for the observed morphospace. Predicting these limits from first principles may facilitate the biological interpretation of the underlying physical constraints. We did not examine cellular-scale dynamics; however, investigating spatial and temporal variations in cell size, shape, and deformation may help clarify how these multiscale processes contribute to size dependent trap closure.

The analysis treats closure as a transition between two static states (open/closed). The dynamics of the curvature change ∂κ/∂t are modeled with a simplistic assumption of constant ϵ _out, which is unlikely to hold true throughout the rapid, non-linear closure process.

Authors’ response:

We thank the reviewer for this comment. Our response to the comment (5) corresponds to answer to this question too.

The model does not account for the role of the midrib, teeth, or the hinge region in guiding and stabilizing closure, which are known to be structurally important.

Authors’ response:

We thank the reviewer for this constructive comment. We added the following statement in the discussion section.

Line 284: the model still did not reflect the complex tissue architecture of the trap such as the midrib, marginal teeth, and hinge region, which may play key roles in guiding and stabilizing trap closure.

The stretching-bending ratio α is adopted from prior work but its direct physical interpretation in the context of the flytrap bilayer, hydrated tissue is not clarified.

Authors’ response:

We thank the reviewer for this important comment. We agree that the physical interpretation of the stretching-bending ratio was not sufficiently clarified in the context of the Venus flytrap.

In general, α represents the relative energetic cost of in-plane stratching versus out-of-plane bending, and is typically determined by geometric and material parameters such as thickness and elastic modulus. In our model, α is adopted as an effective parameter to capture the balance between these two deformation modes at the organ scale.

In the context of the flytrap, which consists of hydrated, multilayered tissues, α can be interpreted as reflecting the mechanical response of a bilayer system in which differential strain – potentially arising from turgor pressure, water transport, and cell wall extensibility – drives bending while being constrained by in-plane stretching resistance. Thus, α effectively incorporates the combined influence of tissue hydration, layer thickness, and cell wall mechanics into a single dimensionless parameter.

We acknowledge that this representation is simplified and does not explicitly account for spatial heterogeneity, viscoelasticity, or fluid-structure coupling. A more rigorous formulation would involve deriving α from measurable physical quantities in a multilayer, poroelastic framework. We have revised the manuscript to clarify this interpretation and to highlight it as an important direction for future work.

Line 153: The dimensionless index α was adopted as an effective parameter to capture the balance between thickness and elastic modulus. In the context of the flytrap, which consists of hydrated, multilayered tissues, α can be interpreted as reflecting the mechanical response of a bilayer system in which differential strain – potentially arising from turgor, water transport, and cell wall extensibility – drives bending while being constrained by in-plane stretching resistance. Thus, α effectively incorporates the combined influence of tissue hydration, layer thickness, and cell wall mechanics into a single dimensionless parameter.

Reviewer: 2

Page 2, line 45: actually, there is one paper published mentioned that only one trigger could initiate the motion of the plant. For your reference: A single touch can provide sufficient mechanical stimulation to trigger Venus flytrap closure.

Authors’ response:

We thank the reviewer for this constructive suggestion. We revised the corresponding part, and added the following statement in the introduction.

Line 59: , and only one touch could initiate the motion of the leaves if

---

## [Decision Letter · Decision Letter 1]

28 Apr 2026

Size–Curvature Constraint in the Closing Motion of Venus Flytrap Leaves

PONE-D-25-65525R1

Dear Dr. Satoru,

We’re pleased to inform you that your manuscript has been judged scientifically suitable for publication and will be formally accepted for publication once it meets all outstanding technical requirements.

Please consider including also the second reviewer's final comments.

Kind regards,

Alice Berardo

Academic Editor

PLOS One

Additional Editor Comments (optional):

Please consider including also the second reviewer's final comments.

Reviewers' comments:

Reviewer's Responses to Questions

**Comments to the Author**

1. If the authors have adequately addressed your comments raised in a previous round of review and you feel that this manuscript is now acceptable for publication, you may indicate that here to bypass the “Comments to the Author” section, enter your conflict of interest statement in the “Confidential to Editor” section, and submit your "Accept" recommendation.

Reviewer #1: All comments have been addressed

Reviewer #2: All comments have been addressed

2. Is the manuscript technically sound, and do the data support the conclusions?

Reviewer #1: Yes

Reviewer #2: Partly

3. Has the statistical analysis been performed appropriately and rigorously? 

Reviewer #1: Yes

Reviewer #2: Yes

4. Have the authors made all data underlying the findings in their manuscript fully available?

Reviewer #1: Yes

Reviewer #2: Yes

5. Is the manuscript presented in an intelligible fashion and written in standard English?

Reviewer #1: Yes

Reviewer #2: Yes

6. Review Comments to the Author

Reviewer #1: The authors have addressed most of my concerns. The reviewer do not has further additional comments for the manuscript.

Reviewer #2: 1. Clarification and consistency of parameter D (important)

In the original geometric formulation proposed by Poppinga and Joyeux, the parameter D is defined as a geometric quantity representing the maximum separation between the two lobes in the reference (closed) configuration, and is used to construct a lifelike initial shape of the trap.Importantly, in that framework, D is treated as a fixed geometric parameter, rather than a dynamic variable describing the temporal evolution of the motion.

In this paper, the parameter D plays a central role throughout the manuscript, as it is used:

1. to distinguish open and closed states (e.g., D = 0 vs D > 0),

2. to describe the deformation process over time,

3. to construct the size–curvature relationship (e.g., D–H morpho space).

Given this central importance, the definition and interpretation of D should be as precise and consistent as possible. At present, D appears to serve multiple roles: a geometric parameter in the surface model (Eq. 1), a descriptor of deformation state, and, in the discussion, a proxy for underlying physiological processes.

While these interpretations are individually reasonable, their combination can be somewhat ambiguous. Please clearly define D in a single, primary sense (e.g., geometric descriptor vs. state variable), and explicitly state how the other interpretations relate to this definition (e.g., whether they are conceptual or derived).

Even a short clarifying paragraph would greatly improve the conceptual clarity of the manuscript.

2. Physical interpretation of D

The added physiological interpretation of D is interesting and provides useful intuition. However, because D is introduced through a geometric model and extracted from morphological fitting, its connection to quantities such as: differential strain, turgor pressure, or ion-mediated processes remains indirect.

To avoid overinterpretation, I suggest: explicitly framing this connection as qualitative or interpretive, unless a direct quantitative relationship is established. This will strengthen the rigor of the manuscript while preserving its interdisciplinary perspective.

7. PLOS authors have the option to publish the peer review history of their article (what does this mean?). If published, this will include your full peer review and any attached files.

Reviewer #1: No

Reviewer #2: **Yes**

---

## [Editor Report · Acceptance letter]

PONE-D-25-65525R1

PLOS One

Dear Dr. Satoru,

I'm pleased to inform you that your manuscript has been deemed suitable for publication in PLOS One. Congratulations! Your manuscript is now being handed over to our production team.

Kind regards,

on behalf of

Dr. Alice Berardo

Academic Editor

PLOS One